# The Influence of Embedded Plasmonic Nanostructures on the Optical Absorption of Perovskite Solar Cells

**Elnaz Ghahremanirad, Saeed Olyaee *, and Maryam Hedayati**

Nano-photonics and Optoelectronics Research Laboratory (NORLab), Shahid Rajaee Teacher Training University, Lavizan 16788-15811, Tehran, Iran; elnaz.ghahremanirad@gmail.com (E.G.); ma.hedayati@srttu.edu (M.H.)

* Correspondence: s_olyaee@srttu.edu, Tel.: +98-21-2297-0030

**Abstract:** The interaction of light with plasmonic nanostructures can induce electric field intensity either around or at the surface of the nanostructures. The enhanced intensity of the electric field can increase the probability of light absorption in the active layer of solar cells. The absorption edge of perovskite solar cells (PSCs), which is almost 800 nm, can be raised to higher wavelengths with the help of plasmonic nanostructures due to their perfect photovoltaic characteristics. We placed plasmonic nanoparticles (NPs) with different radii (20–60 nm) within the bulk of the perovskite solar cell and found that the Au nanoparticles with a radius of 60 nm increased the absorption of the cell by 20% compared to the bare one without Au nanoparticles. By increasing the radius of the nanoparticles, the total absorption of the cell will increase because of the scattering enhancement. The results reveal that the best case is the PSC with the NP radius of 60 nm.

**Keywords:** perovskite solar cell; plasmonic nanoparticles; absorbance; electromagnetic field distribution

## 1. Introduction

One of the problems of methylammonium lead halide (MAPbI$_3$) perovskite solar cells (PSCs) is their low absorption edge, which is around 800 nm. Therefore, they are not able to absorb the near-infrared parts of the solar spectrum. In this regard, different nanostructures have been suggested to enhance light trapping and light absorption in the bulk of the PSC. Plasmonic nanostructures have been used to improve the ability of light absorption in the PSC [1–3].

A considerable improvement has been shown in the light current of MAPbI$_3$ PSCs by including metallic nanoparticles (NPs) with a diameter of 40–80 nm in which (Au, Ag) shells coated the surface of metal oxides (TiO$_2$, SiO$_2$). The interesting point is that in spite of the enhancement in current, there is no improvement in the light absorption [4,5]. At the same time, some research studies have been done on the size, shape, and combination of NPs as absorption enhancer in the PSCs. In [6], the frequency domain time domain (FDTD) method was used to simulate a three-dimensional PSC structure and the absorption enhancement of 6–12% was obtained depending on the thickness of the absorber layer. The results were calculated at the occupation volume of 10%, which shows the volume of the NPs in proportion to the volume of the absorber layer. They neglected the negative effects of the number of NPs on the carrier transport. Dabirian et al. created an effective scattering in the absorber layer with low absorption loss by using core-shell NPs with the dimension comparable to the wavelength in which a thin metallic shell coated the surface of a dielectric core. These NPs were introduced as an efficient light trapping structure inside the dye-sensitized solar cells with the thickness of multiple micrometers [7].

Having caused the enhancement of absorption and efficiency of PSCs, the core-shell NPs have been the major subject of many researchers. In a mesoscopic PSC structure which used Au NPs inside the meso TiO₂ layer, the absorption decreased compared to the structure without the NPs. The absorption as well as the value of open-circuit voltage and short-circuit current increased as a result of using the MgO shell as a passive layer on the surface of Au NPs. The reason for this improvement was the increase in carrier transport and the decrease in recombination loss [8]. Moreover, these core-shell NPs improved the stability of the PSCs. In addition, other works have suggested the use of SiO₂ as a shell for Au NPs. These nanostructures incorporated as either nanorods or nanospheres in the PSCs. The goal of all the research studies was the enhancement in the absorption of incident light in addition to the output power efficiency of the cell [9–12]. These NPs were inserted both inside the absorber layer [9,10] and on the interface of the absorber and electron transporting layers [11]. Furthermore, the plasmonic NPs were used inside the PSCs with inorganic transporting layers. It was shown in reference [13] that plasmonic NPs at the interface of perovskite and NiO layers could excite plasmon resonance with the wavelength range of 650–800 nm due to strong scattering and enhanced electric field around the NPs. The main reason for using plasmonic NPs is the lower absorption of thin film perovskite layers [14].

In another study, a designed structure of solar cell showed an approximate 5% improvement in performance, based on the embedding of Au@TiO₂ nanorods (NRs) into the TiO₂ layer. The optimized device exhibited 20.1% efficiency, which is by far the highest efficiency based on the incorporation of Au@TiO₂ NRs into the planar PSC [15]. Recently, Wang et al. successfully synthesized Ag@SiO₂ NPs with a modified Stöber method and developed PSCs with different contents of Ag@SiO2 NPs. The results revealed that by using an optimal content of NPs, a 19.46% improvement of the power conversion efficiency was obtained [16]. In addition, the successful integration of plasmonic gold nanostars into mesoporous TiO₂ photoelectrodes for PSCs has been reported. This structure exhibited a device efficiency of up to 17.72% [17].

The Au material has chemical stability in many environments and it is a suitable material in localized surface plasmon resonance (LSPR) applications. However, the easy oxidation of Ag upon air exposure can change its plasmonic properties which is why we used Au NPs in a simulation of the device [18]. In this work, we considered the influence of the number, position, volume, and spacing of the Au NPs from each other. First, we put one Au NP with various radii at the center of the absorber layer and then we changed the position of the NP with the best radius in the direction of the light source. Third, we put two Au NPs inside the absorber layer and compared it with the PSC with only one NP. In this paper, we show that the internal absorption of the perovskite solar cell using plasmon nanoparticles is increased. The rest of the paper is categorized as follows: a description of the perovskite solar cell model is presented in Section 2. Then, the simulation results are described in Section 3. Finally, a conclusion is drawn in Section 4.

## 2. Materials and Methods

The schematic of the device is illustrated in Figure 1. The PSC was irradiated in the z-direction at the normal angle under the unpolarized AM1.5G spectrum at one sun intensity. The boundary conditions were periodic in the x- and y-directions. Furthermore, we applied the boundary condition of perfectly matched layers (PMLs) in the z-direction. The thickness of the perovskite, TiO₂, and Spiro-OMeTAD layers were 200 nm, 100 nm, and 80 nm, respectively. We extracted the refractive index $n(\lambda)$ and extinction coefficient $k(\lambda)$ of these materials for optical simulation [19,20].

A part of the perovskite was removed and replaced with Au. By drawing the refractive index curve, it can be proved that the section represents Au. To simulate optical properties of the PSC, we have to calculate the total optical absorption. It is computed as follows [18,21–22]:

$$P_{abs}(\lambda) = \int \varepsilon_\circ \omega(\lambda) |E(x,y,z,\lambda)|^2 n(\lambda) k(\lambda) \, dV , \tag{1}$$

where ω is the angular frequency, *E* is the electric field intensity, and ε₀ is the vacuum permittivity. The optical current density J$_L$ is calculated by integrating the total optical absorption P$_{abs}$($\lambda$) over the wavelength range of 400–850 nm as:

$$J_L = \frac{q}{hc} \int_{400\,nm}^{850\,nm} \lambda\, P_{abs}(\lambda)\, I(\lambda)\, d\lambda, \tag{2}$$

where *q* is the elementary charge, *h* is the Planck constant, *c* is the speed of light, and *I*($\lambda$) is the intensity of the AM1.5G solar spectrum.

## 3. Results

We placed metallic nanoparticles inside the absorber layer to utilize the localized surface plasmon effect so as to enhance the electric field intensity around the NPs. The situation of the NPs in the layer as well as the number of the NPs have a crucial effect on the absorption of the solar cell. First, we put different NPs with various radii inside the active layer, the influence of which on the absorbance spectra of the solar cell has been explored. Figure 2 shows the absorbance spectra of the perovskite layer with different Au nanoparticle radii for different NPs (20–60 nm). As the nanoparticle radius increases, absorption increases.

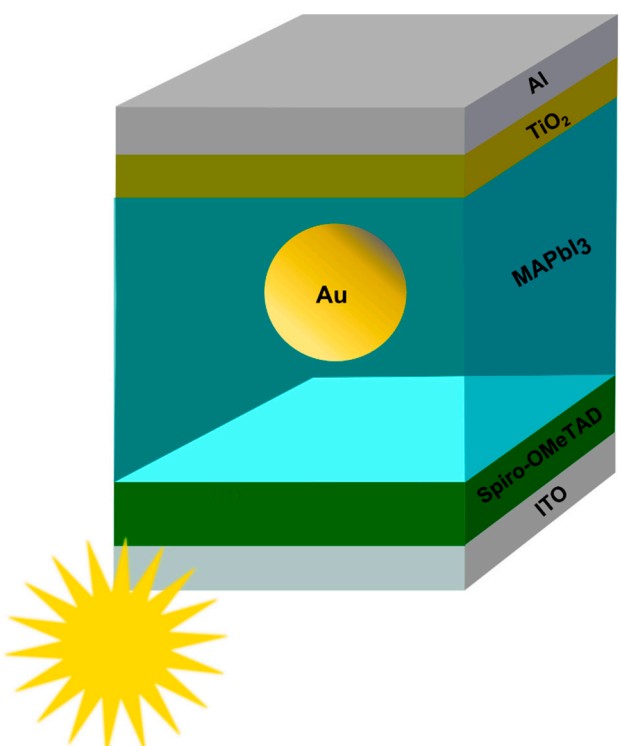

**Figure 1.** The schematic of the perovskite solar cell with a Au nanoparticle (NP).

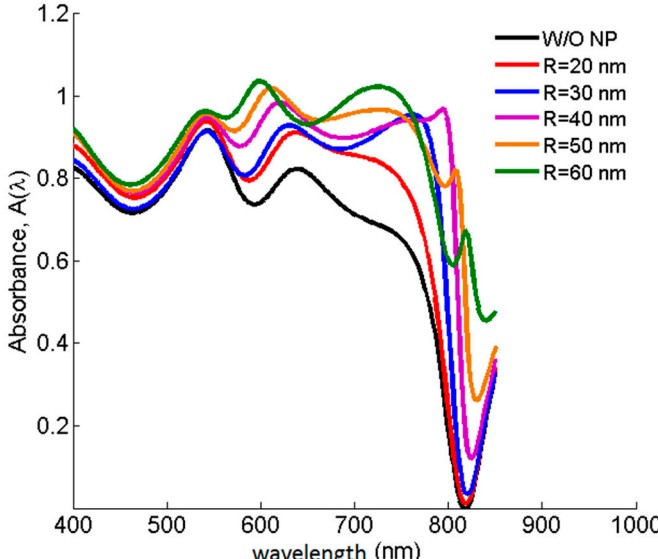

**Figure 2.** The absorbance spectra of the perovskite layer with different Au nanoparticle radii (the black curve shows the absorbance spectrum of the reference cell).

By increasing the radius of the nanoparticles, the total absorption of the cell increases as a result of the scattering enhancement. Figure 3 shows the absorption enhancement factor for different NP radii with respect to the reference cell. By placing nanoparticles, the absorption around the wavelength of 800 nm (where perovskite without nanoparticles has zero absorption) has increased. Obviously, there is a direct correlation between the nanoparticle radii and the absorption, meaning the absorption rises by choosing larger nanoparticles, although a nonlinearity appears when the Au nanoparticles have a radius of 30 nm.

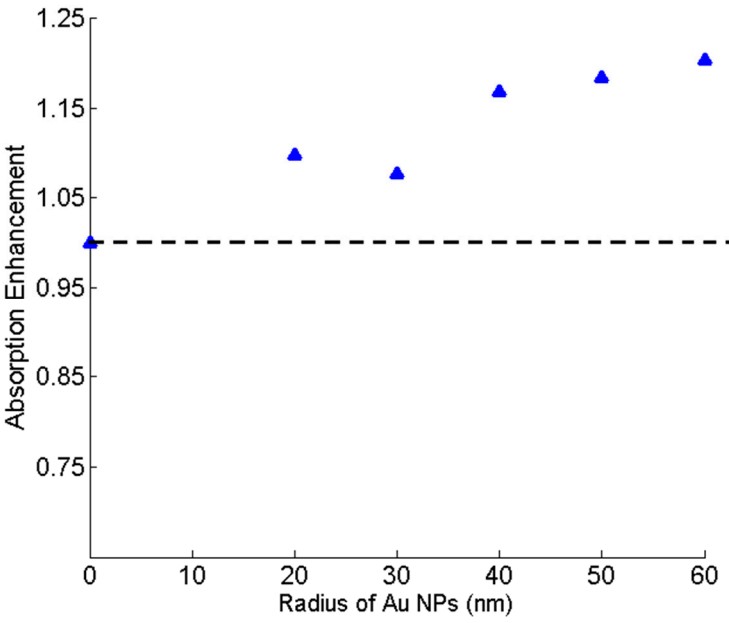

**Figure 3.** The absorption enhancement factor of the perovskite solar cell (PSC) including a plasmonic NP with different radii.

It is clear that the best case is the PSC with the NP radius of 60 nm. In this case, the total absorption increases by 20% compared to the bare PSC without any NP. Next, we changed the position of the Au NP with a radius of 60 nm in the z-direction. Initially, the NP was placed very

close to the unpolarized light source or Spiro-OMeTAD layer. Later, the NP was located at the top of the active layer or near the TiO₂ layer. Figure 4a shows the absorbance spectra of the PSC including an NP with a radius of 60 nm in different situations.

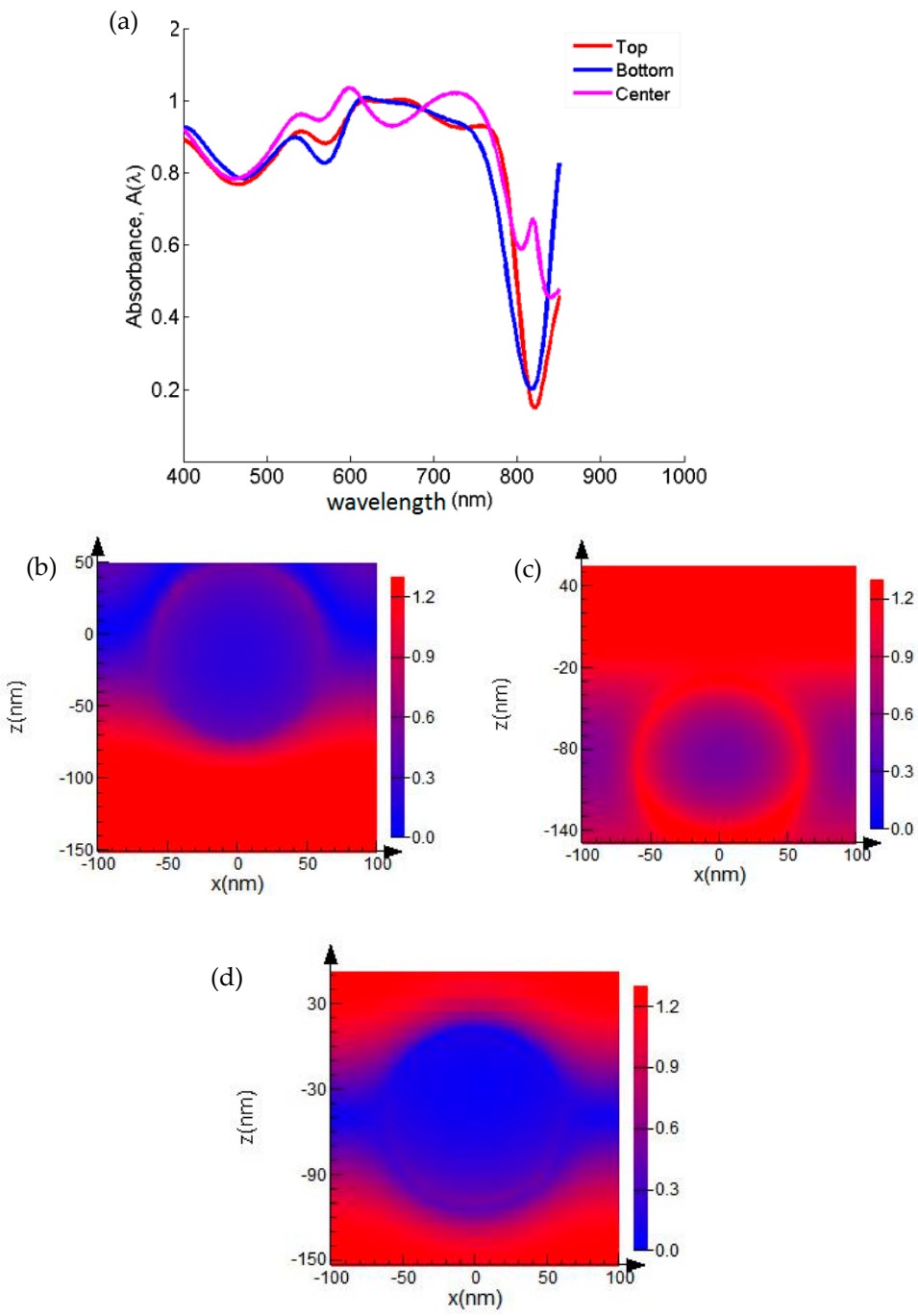

**Figure 4.** (**a**) The absorbance spectra of the PSC including an NP with a radius of 60 nm situated in different spots in the direction of the light source. (**b**–**d**) The electric field distribution spectra related to the locations of top, bottom, and center of the absorber layer.

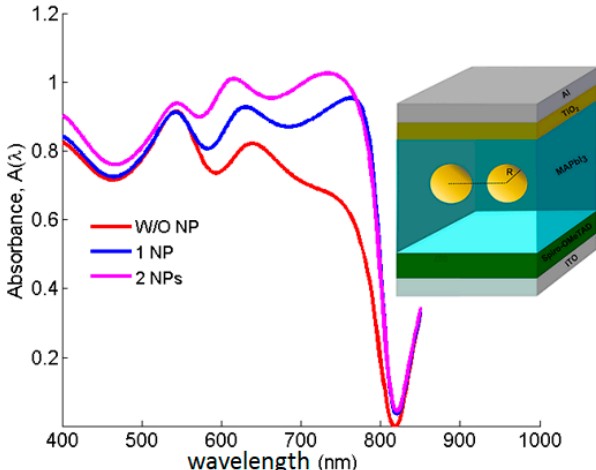

**Figure 5.** The comparison between the absorption spectra of the perovskite solar cell without the NP and including one and two Au NPs with a radius of 30 nm (inset shows the schematic of the PSC with two embedded Au NPs).

The electric field distribution spectra of all the situations are shown in Figure 4b–d for top, bottom, and center positions, respectively. According to the figures, when the NP is in the center of the absorber layer, it absorbs more light with respect to other situations, as can be deduced from the absorption spectra.

Afterward, two Au NPs were put inside the active layer and the results were compared to the case where there was only one Au NP as well as the case without any NPs. A radius of 30 nm was considered for these NPs.

In Figure 5, the absorbance spectra are plotted for three different cases. Using more NPs makes the light absorption magnitude escalate in the visible region. The constant volume of the unit cell was also considered in our calculations. Therefore, there is no more space to insert larger NPs or more NPs. The results effectively show the positive influence of the metallic NPs on the total absorption of the PSC. Moreover, their effect on the device performance can be explored in future research studies.

As a brief comparison, plasmonics is used to increase the absorption of infrared frequencies in thin film perovskite solar cells in [23]. The problem of transmission losses in the band of 82–950 nm is theoretically solved by embedding gold nanoparticles into the bottom part of the perovskite layer. Nanoparticles located there increase the useful absorption in the infrared spectral hole of the perovskite by 32% and do not distort the photovoltaic absorption at other wavelengths. However, by inserting the nanoparticles, we concluded that the near field around the nanoparticles could increase the light harvest. In addition, by increasing the radius of the nanoparticle, the amount of light scattering increased, which is also shown in [24].

## 4. Conclusions

Perovskite solar cells have aroused intense interest in recent years. Incorporating nanostructures into solar cell layers can enhance their performance to harvest more energy. We put plasmonic nanospheres inside the device, whereby the enhanced electric field around the nanoparticles increased the possibility of light absorption inside the active layer. The influence of different nanoparticle radii as well as the number of the nanoparticles on the absorbance spectra were discussed and compared with the bare perovskite solar cell without nanoparticles. The results showed an improvement in the magnitude and width of the absorbance spectra related to the reference perovskite solar cell. The perovskite solar cell is composed of a Au NP with a radius of 60 nm which is able to increase the total absorption of the bare perovskite solar cell by 20%. As a future work, the experimental results of this cell will be considered at the next stage.

**Author Contributions:** E. Ghahremanirad designed and performed simulations, analyzed data, and drafted the final manuscript. S. Olyaee and M. Hedayati edited and prepared the final draft of the manuscript.

**Funding:** This work was supported by Shahid Rajaee Teacher Training University (SRTTU) under contract number 23106.

**Acknowledgments:** This research was carried out in the Nano-photonics and Optoelectronics Research Laboratory (NORLab) and the authors would like to thank the head of the NORLab for the kind help provided.

**Conflicts of Interest:** The authors declare no conflict of interest.

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
