# Peer review of "The Influence of Embedded Plasmonic Nanostructures on the Optical Absorption of Perovskite Solar Cells"

_photonics, doi:10.3390/photonics6020037_

Round 1
Reviewer 1 Report
The authors have successfully calculated the enhancement of absorption in plasmonic perovskite solar cells by using FDTD simulations. The enhancement of absorption is obtained around 800 nm wavelength which is the band edge of methylammonium lead halide perovskite, which is corresponding to the previous experimental studies. It is optimized that the radius of 60 nm is appropriate size for the light trapping in the perovskite solar cells based on the highly scattering cross-section. The calculated results are useful for the design of plasmonic perovskite solar cells. Therefore, I judged that the manuscript can be publishable after a minor revision.
1. The best performance was obtained by the Au nanoparticles with a radius of 60 nm. However, the authors did not show the simulated data with a radius of 70 nm or 80 nm. Therefore, it cannot be concluded that the best radius is 60 nm. The authors should consider about it.
2. Why did the authors not simulate the density dependence of Au nanoparticles by using periodic boundary conditions on FDTD simulations? The authors only simulated one and two particles without a discussion of spacing between Au nanoparticles.
3. For the presentation of near-field intensity distribution, the polarization conditions and its direction if it is linear polarized conditions should be explained.
4. In this manuscript, there are “nano particle” and “nanoparticle” as a description. It should be unified as “nanoparticle”.
Author Response
Cover Letter
23 March 2019
The Influence of Embedded Plasmonic Nanostructures on Optical Absorption of Perovskite Solar Cells
Photonics
Editor-in-Chief
Dear Professor Nelson Tansu
I deeply appreciate the attention you have given the paper. We carefully studied the reviewer's comments and found them very important and useful to improve the paper. We would like to thank you and reviewers for valuable and useful comments and recommendations. In meeting the comments, we revised the paper and we hope the freshly revised version satisfies the comment and recommendations. All corrections are highlighted in Yellow in the revised paper. However, we would like to make the following points regarding the reviewer's comments.
Yours sincerely
Prof. Saeed Olyaee
Nano-photonics and Optoelectronics Research Laboratory (NORLab),
Faculty of Electrical Engineering,
Shahid Rajaee Teacher Training University (SRTTU),
Lavizan, 16788-15811, Tehran, Iran
E-mail: s_olyaee@sru.ac.ir
Reviewer Comments:
Reviewer #1
The authors have successfully calculated the enhancement of absorption in plasmonic perovskite solar cells by using FDTD simulations. The enhancement of absorption is obtained around 800 nm wavelength which is the band edge of methylammonium lead halide perovskite, which is corresponding to the previous experimental studies. It is optimized that the radius of 60 nm is appropriate size for the light trapping in the perovskite solar cells based on the highly scattering cross-section. The calculated results are useful for the design of plasmonic perovskite solar cells. Therefore, I judged that the manuscript can be publishable after a minor revision.
Comment:
1. The best performance was obtained by the Au nanoparticles with a radius of 60 nm. However, the authors did not show the simulated data with a radius of 70 nm or 80 nm. Therefore, it cannot be concluded that the best radius is 60 nm. The authors should consider about it.
Response:
Thank you very much for this comment. The dimension of the unit cell does not allow for considering higher radii more than 60 nm. Therefore, we just considered the values which are the proper fit for the unit cell dimension. Therefore, the structure is simulated for 20-60 nm radii as shown in Fig. 2 which the results are compared with reference cell.
Comment:
2. Why did the authors not simulate the density dependence of Au nanoparticles by using periodic boundary conditions on FDTD simulations? The authors only simulated one and two particles without a discussion of spacing between Au nanoparticles.
Response:
We used periodic boundary conditions in both x and y directions in the FDTD simulation (please see section 2 in page 2).
Comment:
3. For the presentation of near-field intensity distribution, the polarization conditions and its direction if it is linear polarized conditions should be explained.
Response:
We used unpolarized light source in the presentation of near-field intensity enhancement. This comment is considered in the last paragraph of page 2 and the first paragraph of page 5.
Reviewer #2
Ghahremanirad et al. explored the effect of plasmonic nanostrcutures in perovskite solar cells. The manuscript reports some interesting results which would be of great interest to the readers of Photonics. Only the concern in the paper is that it is not very clear this paper is based on experimental investigation or theoretical study. The authors should make this very clear in both Abstract and Introduction. Other minor suggestions include:
Comment:
Figure 1: the authors should draw the device structure in the other way around (e.g. Conductive electrode/ETM/Perovskite/HTM/Electrode).
Response:
In meeting the comment, we revised the solar cell structure as shown in Fig. 1 (please see page 3). We hope the revised figure satisfies the reviewer’ comment.
Comment:
in Figure 2, Figure 4a, and Figure 5, X-axis should be wavelength, instead of lambda.
Response:
The X-axis of all Figs. 2, 4a, and 5 are revised (please see pages 4-6).
Comment:
The authors should include some important advances that have been made in this field recently:
- J. Mater. Chem. A, 2017, 5, 12034.
- ChemSuSChem, 2017, 10, 3750-3753.
- Sol. RRL 2018, 2, 1800061.
Response:
Thank you very much for this comment. We add references #12, 15-17. The corrections are highlighted in Yellow in the revised paper.

Reviewer 2 Report
Ghahremanirad et al. explored the effect of plasmonic nanostrcutures in perovskite solar cells. The manuscript reports some interesting results which would be of great interest to the readers of Photonics. Only the concern in the paper is that it is not very clear this paper is based on experimental investigation or theoretical study. The authors should make this very clear in both Abstract and Introduction. Other minor suggestions include:
Figure 1: the authors should draw the device structure in the other way around (e.g. Conductive electrode/ETM/Perovskite/HTM/Electrode).
in Figure 2, Figure 4a, and Figure 5, X-axis should be wavelength, instead of lambda.
The authors should include some important advances that have been made in this field recently:
- J. Mater. Chem. A, 2017, 5, 12034.
- ChemSuSChem, 2017, 10, 3750-3753.
- Sol. RRL 2018, 2, 1800061.
Author Response

(The authors gave the same response as above.)
